# Pimasertib Versus Dacarbazine in Patients With Unresectable *NRAS*-Mutated Cutaneous Melanoma: Phase II, Randomized, Controlled Trial with Crossover

**DOI:** 10.3390/cancers12071727

**Published:** 2020-06-29

**Authors:** Celeste Lebbé, Caroline Dutriaux, Thierry Lesimple, Willem Kruit, Joseph Kerger, Luc Thomas, Bernard Guillot, Filippo de Braud, Claus Garbe, Jean-Jacques Grob, Carmen Loquai, Virginia Ferraresi, Caroline Robert, Paul Vasey, Robert Conry, Richard Isaacs, Enrique Espinosa, Armin Schueler, Giorgio Massimini, Brigitte Dréno

**Affiliations:** 1AP-HP Dermatology CIC Department, Saint Louis Hospital, and INSERM U976, Université de Paris, 75010 Paris, France; 2Dermatology, Hopital Saint-Andre—CHU, 33000 Bordeaux, France; caroline.dutriaux@chu-bordeaux.fr; 3Medical Oncology Department, Comprehensive Cancer Center Eugène Marquis, 35000 Rennes, France; t.lesimple@rennes.unicancer.fr; 4Internal Oncology, Erasmus MC Cancer Institute, 3008 AE Rotterdam, The Netherlands; f2hkruitwhj@kpnmail.nl; 5Medical Oncology, Institut Jules Bordet, 1000 Brussels, Belgium; joseph.kerger@bordet.be; 6Department of Dermatology, Centre Hospitalier Lyon Sud, 69310 Pierre Bénite, France; luc.thomas@chu-lyon.fr; 7Department of Dermatology, Hôpital Saint Eloi, 34295 Montpellier, France; docbernardguillot@gmail.com; 8Department of Medical Oncology, Istituto Nazionale dei Tumori, Università degli Studi di Milano, 20133 Milano, Italy; filippo.debraud@istitutotumori.mi.it; 9Department of Dermatology, University Hospital Tübingen, 72076 Tübingen, Germany; claus.garbe@med.uni-tuebingen.de; 10Department of Dermatology and Cutaneous Oncology Service, Hôpital de la Timone, 13005 Marseille, France; jean-jacques.grob@ap-hm.fr; 11Department of Dermatology, University Medical Center Mainz, 55019 Mainz, Germany; carmen.loquai@unimedizin-mainz.de; 12Division of Medical Oncology 1, IRCCS “Regina Elena” National Cancer Institute, 00144 Roma, Italy; virginia.ferraresi@ifo.gov.it; 13Dermatology Department, Institut Gustave Roussy and Paris Sud University, 94800 Villejuif, France; Caroline.ROBERT@gustaveroussy.fr; 14Icon Cancer Care, The Wesley Hospital, Auchenflower, QLD 4066, Australia; PVasey@iconcancercare.com.au; 15Comprehensive Cancer Center, University of Alabama at Birmingham, Birmingham, AL 35233, USA; rconry@uabmc.edu; 16MidCentral Regional Cancer Treatment Service, Palmerston North Hospital, Palmerston North 4442, New Zealand; richard.isaacs@midcentraldhb.govt.nz; 17Medical Oncology Department, Hospital Universitario La Paz, 28046 Madrid, Spain; eespinosa00@hotmail.com; 18Global Biostatistics Oncology, Merck KGaA, 64293 Darmstadt, Germany; Armin.Schueler@merckgroup.com; 19GCDU Oncology, Merck KGaA, 64293 Darmstadt, Germany; Giorgio.Massimini@merckgroup.com; 20Department of Dermato Cancerology, CIC 1413, CRCINA Inserm 1232, CHU Nantes, 44093 Nantes, France; brigitte.dreno@atlanmed.fr

**Keywords:** malignant melanoma, dacarbazine, pimasertib, N-(2,3-dihydroxypropyl)-1-((2-fluoro-4-iodophenyl)amino)isonicotinamide, progression-free survival, quality of life, adverse events

## Abstract

This study investigated the efficacy and safety of pimasertib (*MEK*1/*MEK*2 inhibitor) versus dacarbazine (DTIC) in patients with untreated *NRAS*-mutated melanoma. Phase II, multicenter, open-label trial. Patients with unresectable, stage IIIc/IVM1 *NRAS*-mutated cutaneous melanoma were randomized 2:1 to pimasertib (60 mg; oral twice-daily) or DTIC (1000 mg/m^2^; intravenously) on Day 1 of each 21-day cycle. Patients progressing on DTIC could crossover to pimasertib. Primary endpoint: investigator-assessed progression-free survival (PFS); secondary endpoints: overall survival (OS), objective response rate (ORR), quality of life (QoL), and safety. Overall, 194 patients were randomized (pimasertib *n* = 130, DTIC *n* = 64), and 191 received treatment (pimasertib *n* = 130, DTIC *n* = 61). PFS was significantly improved with pimasertib versus DTIC (median 13 versus 7 weeks, respectively; hazard ratio (HR) 0.59, 95% confidence interval (CI) 0.42–0.83; *p* = 0.0022). ORR was improved with pimasertib (odds ratio 2.24, 95% CI 1.00–4.98; *p* = 0.0453). OS was similar between treatments (median 9 versus 11 months, respectively; HR 0.89, 95% CI 0.61–1.30); 64% of patients receiving DTIC crossed over to pimasertib. Serious adverse events (AEs) were more frequent for pimasertib (57%) than DTIC (20%). The most common treatment-emergent AEs were diarrhea (82%) and blood creatine phosphokinase (CPK) increase (68%) for pimasertib, and nausea (41%) and fatigue (38%) for DTIC. Most frequent grade ≥3 AEs were CPK increase (34%) for pimasertib and neutropenia (15%) for DTIC. Mean QoL scores (baseline and last assessment) were similar between treatments. Pimasertib has activity in *NRAS*-mutated cutaneous melanoma and a safety profile consistent with known toxicities of *MEK* inhibitors. Trial registration: ClinicalTrials.gov, NCT01693068.

## 1. Introduction

Dacarbazine (DTIC) was the standard of care (SOC) for metastatic melanoma [1] until agents, such as ipilimumab, nivolumab, and pembrolizumab [2,3,4,5], as well as the oncolytic virus therapy talimogene laherparepvec [6], were approved. *BRAF* and *NRAS* mutations occur in approximately 40% and 15–20% of cutaneous melanomas, respectively [7,8]. For *BRAF*-mutated metastatic melanoma, targeted therapies—such as the *BRAF* inhibitors dabrafenib, vemurafenib, and encorafenib—are available, which in combination with *MEK* inhibitors—such as trametinib, cobimetinib, and binimetinib—can help overcome acquired resistance and improve survival and response outcomes [9,10,11,12]. For the more aggressive *NRAS*-mutated melanomas [13,14], however, there are no specifically targeted therapies available; thus, a significant unmet need remains for new treatment options.

Pimasertib is an orally bioavailable, selective small-molecule *MEK1/2* inhibitor. At the recommended phase II dose (RP2D) of 60 mg twice-daily (bid) it has an acceptable safety profile in patients with solid tumors and potential efficacy in patients with *BRAF*- and/or *NRAS*-mutated melanoma tumors [15].

This phase II study (NCT01693068) aimed to confirm the efficacy and safety of pimasertib at the RP2D in patients with *NRAS*-mutated cutaneous melanoma, and to compare this with DTIC, the SOC at the time of study initiation.

## 2. Results

### 2.1. Patient Population

One hundred and ninety-four patients were randomized (ITT set; DTIC *n* = 64, pimasertib *n* = 130) and 191 received treatment (safety population; DTIC *n* = 61, pimasertib *n* = 130) (Appendix A). The most common reasons for discontinuation were PD (118 (62%) patients) and TEAEs (69 (36%) patients). At data cut-off, six (3%) patients were still on treatment (DTIC *n* = 1, crossover *n* = 1, pimasertib *n* = 4).

Baseline characteristics (ITT population) were balanced between the DTIC and pimasertib arms (Table 1; Appendix A); most patients had an Eastern Cooperative Oncology Group performance (ECOG PS) of 0 (69% versus 69%, respectively), with a similar percentage of patients with M1c melanoma (67% versus 64%, respectively) and lactate dehydrogenase >upper limit of normal (36% versus 42%, respectively).

### 2.2. Efficacy

The primary objective of the trial was achieved; progression-free survival (PFS) was significantly improved in the pimasertib arm compared with the DTIC arm (median 13 versus 7 weeks, respectively; hazard ratio (HR) 0.59, 95% confidence interval (CI) 0.42–0.83; P = 0.0022) (Figure 1a). Six-month PFS rates were 17% versus 9%, respectively. Independently assessed PFS supported investigator-assessed PFS outcomes (Figure 1b). Sensitivity analyses including all deaths and all scans supported the investigator-assessed PFS findings (Appendix A), although this was not statistically significant when based on independent assessment (P = 0.1454). Subgroup analyses indicated a consistent trend in favor of pimasertib treatment (Figure 1c; Appendix A).

No difference in OS was observed between patients receiving pimasertib and DTIC (median 9 months versus 11 months, respectively; HR 0.89, 95% CI 0.61–1.30) (Appendix A) and no prognostic factors were identified (Appendix A); notably, 64% of patients in the DTIC arm crossed over to pimasertib.

Best overall response data suggested a consistent benefit with pimasertib versus DTIC (Table 2). Investigator-evaluated ORR was 27% for pimasertib and 14% for DTIC (odds ratio 2.24, 95% CI 1.00–4.98, *p* = 0.0453). Similarly, disease control rate (DCR) was greater for patients receiving pimasertib versus DTIC (odds ratio 2.65, 95% CI 1.23–5.69, *p* = 0.0106). Independently assessed ORR and DCR data supported these findings, with a trend toward a greater benefit with pimasertib (Table 2).

### 2.3. Safety

Patients remained on treatment for a median of 10 versus 7 weeks in the pimasertib and DTIC arms, respectively, and 9 weeks in the crossover group.

The frequency of TEAEs was similar between the arms (100% for pimasertib and crossover versus 98% for DTIC). All patients in the pimasertib and crossover arms and 89% of patients in the DTIC arm had ≥1 treatment-related TEAE.

Serious adverse events (SAEs) (57%, 63%, and 20%) and treatment-related SAEs (45%, 49%, and 7%) were more frequent in the pimasertib and crossover groups than in the DTIC arm, respectively; this was also observed for TEAEs leading to treatment modification (81%, 78%, and 26%) and permanent treatment discontinuations (47%, 39%, and 5%) (Appendix A). TEAEs were the primary reason for death in five patients (8%) receiving DTIC and one patient (1%) receiving pimasertib.

The most common TEAEs in the pimasertib and crossover arms were diarrhea (82% and 76%, respectively) and blood CPK increase (68% and 71%, respectively), and, in the DTIC arm, were nausea (41%) and fatigue (38%) (Appendix A). More patients receiving pimasertib versus DTIC had grade ≥3 TEAEs (85% versus 41%, respectively; Table 3) and grade ≥3 treatment-related TEAEs (77% versus 28%, respectively). The most frequent treatment-related grade ≥3 TEAEs were increased CPK for pimasertib (34%) and neutropenia (15%) and thrombocytopenia (13%) for DTIC.

Ocular events (SRDs) occurred in 58%, 51%, and 31% of patients in the pimasertib, crossover and DTIC groups, respectively; 4%, 5% and 0% of patients, respectively, had retinal vein occlusion (RVO) events (Appendix A). Ocular AEs of special interest (AESIs) were largely reversible (Appendix A), mild-to-moderate in intensity and had no negative impact on visual acuity.

CPK increases occurred in 57%, 59%, and 28% of patients in the pimasertib, crossover and DTIC groups, respectively, which led to treatment delay/interruption in 31%, 32%, and 0%, and to discontinuation in 13%, 2%, and 0% of patients, respectively. CPK was reversible in 95%, 92%, and 88% of patients, respectively; eight patients had mild-to-moderate muscle pain, otherwise cases were asymptomatic.

At baseline, four patients in the pimasertib arm had a left ventricular ejection fraction (LVEF) <50%; the greatest on-treatment LVEF reduction in these patients was <15%. Of patients with baseline LVEF ≥50% (pimasertib *n* = 115, crossover *n* =37), 13 (pimasertib *n* = 10, crossover *n* = 3) had a decrease to <45% while on treatment. However, of these 13 patients, most had strong confounding risk factors for LVEF alteration, such as elderly age (65–75 years; *n* = 8), hypertension (*n* = 9), decreased hemoglobin (*n* = 8), and aortic stenosis (*n* = 1). LVEF alteration led to treatment modification in three patients receiving pimasertib (2%). LVEF was not assessed in patients receiving DTIC for comparison.

No new safety signals for pimasertib were identified.

### 2.4. Quality of Life

Mean QoL scores were similar between treatment arms. Mean (standard deviation) Functional Assessment of Cancer Therapy (FACT)–General total scores were: 80 (16) and 75 (15) at baseline, and 74 (17) and 70 (15) at end of treatment, for pimasertib and DTIC, respectively (Appendix A).

## 3. Discussion

This study demonstrated a significant improvement in PFS with pimasertib at the RP2D, compared with DTIC, in patients with *NRAS*-mutated cutaneous melanoma, whether assessed by investigators or independently. The phase III NEMO study observed a similar PFS improvement with the *MEK* inhibitor binimetinib versus DTIC (HR 0.62; *p* < 0.001) in treatment-naïve or immunotherapy-pretreated patients with *NRAS*-mutant metastatic melanoma [16].

Clinically relevant benefits were observed for tumor response with pimasertib over DTIC. This improvement was less pronounced with independent analysis, likely due to differences in PR and SD estimates between these assessments. Notably, a considerable number of patients randomized to DTIC crossed over to pimasertib and may have crossed over too early, which may have influenced not only response outcomes but also OS estimates. Nonetheless, the higher ORR favors pimasertib over DTIC.

The safety profile of pimasertib was consistent with previous studies [15,16,17,18,19] and known *MEK* inhibitor class effects [16,20,21], with no new safety signals and no impact on QoL. As expected, ocular AESI or CPK increases were seen more frequently in patients who received pimasertib than DTIC; however, the higher than expected incidence of ocular TEAEs with DTIC may reflect the thorough evaluation of ocular toxicity conducted in this study. Although some patients experienced an LVEF decrease while receiving pimasertib, most of these patients had strong confounding factors, such as hypertension, elderly age (≥65 years), and valvular heart disease (aortic stenosis), which are considered major clinical factors for LVEF alteration [22]. Notably, patients were treated for longer with pimasertib than with DTIC and the stringent rules for dose modifications may have resulted in more frequent de-escalation of pimasertib treatment, which is likely reflected in the higher incidence of discontinuation/modification of treatment due to TEAEs with pimasertib. For example, for CPK assessments, treatment discontinuation was based upon laboratory findings/non-resolution, with no integration of further symptomatic findings. However, although dose modifications and interruptions may have affected the overall pimasertib exposure, pimasertib still provided greater efficacy than DTIC.

One limitation of this study is that the comparator, DTIC, is no longer the SOC for metastatic melanoma and has been superseded by targeted and immune therapies [23]. In addition, permanent treatment discontinuations may have affected pimasertib exposure, which, along with the extensive crossover from DTIC to pimasertib, is likely to have affected safety and survival outcomes. Indeed, *PD-1* blockade either alone or combined with anti-*CLTA-4* as first-line therapy is now considered to be the standard of care of advanced *NRAS*-mutated melanoma. Anti *PD1* alone leads to a best ORR around 40% and a median PFS up to 8 months, while the combination regimen allows for best ORR and a median PFS of 57% and 11.5 months, respectively [24,25].

In conclusion Pimasertib has shown clinical activity in patients with *NRAS*-mutated cutaneous melanoma and a safety profile that is consistent with the known toxicities of *MEK* inhibitors. These findings support the potential combination of pimasertib with other agents, such as those targeting the *PI3K*/*mTOR* pathway or other *MEK* pathway components, or immune checkpoint inhibitors.

## 4. Materials and Methods

### 4.1. Study Design and Treatment

This phase II, multicenter, randomized, open-label trial was conducted in compliance with the principles of the Declaration of Helsinki, the International Council for Harmonization Note for Guidance on Good Clinical Practice (ICH topic E6, 1996) and all applicable regulatory requirements. this research has been approved by the Quorum Review IRB on 07/03/12 (ethic code: NCT01693068).

Patients were randomized 2:1 (permuted block randomization using an interactive voice response system performed centrally by PPD under the supervision of the Sponsor) to receive pimasertib (60 mg; oral bid) or DTIC (1000 mg/m^2^; intravenously) on Day 1 of each 21-day cycle (Appendix A). Randomization was stratified by Eastern Cooperative Oncology Group performance status (ECOG PS; 0 versus 1). Patients progressing on DTIC could switch to pimasertib (crossover) if they met the eligibility criteria. After progressive disease (PD), patients were followed up every 6 months to assess survival.

### 4.2. Patients

Eligible patients were aged ≥18 years and had histologically or cytologically confirmed, unresectable locally advanced or metastatic cutaneous melanoma (stage IIIc or IV (M1a-c)) with a confirmed *NRAS* mutation, but no prior systemic treatment for locally advanced/metastatic melanoma (see Appendix A for full inclusion and exclusion criteria, including protocol amendments).

The study report states the following: For confirmation of entry criteria, centralized testing of *N-RAS* mutation status was to be done on a tumor sample using the Sanger technique. If the *N-RAS* status was already known at time of inclusion in the trial, the method used for testing and the date were also to be recorded. *N-RAS* status had to be known prior to performing trial specific screening assessments and prior to randomization in the trial. Analysis of the presence of different types of genetic variants (e.g., gene mutations, copy number variations, methylations), expression levels pathway associated proteins, and marker of proliferation were to be performed on fresh tumor tissue biopsies or archived tumor tissue.

Patients were recruited from 88 centers in Australia, Europe, Israel, New Zealand, and the USA, from December 2012 to July 2014.

### 4.3. Assessments

Tumor response was assessed according to Response Evaluation Criteria In Solid Tumors (RECIST, version 1.0), at baseline, Day 1 of Cycles 3, 5, 7, 9, 11, and 13, and every 4 cycles thereafter. Progression-free survival (PFS) was assessed by investigators, and supported by blinded, retrospective, independent central review of all available imaging data. PFS (weeks) was defined as: ([{date of first PD or death or date of censoring} − date of randomization] + 1) / 7. An event was PD or death if it did not occur after 2 or more missed tumor assessments. In patients with only a date of disease progression date or death after 2 or more missed tumor assessments, the PFS time was censored on the minimum of the date of the last tumor assessment and the analysis cut-off date. Median number of weeks rounded to nearest whole number.

Patient reported quality of life (QoL) was assessed by Functional Assessment of Cancer Therapy (FACT)–Melanoma from baseline to last assessment (prior to objective PD).

Safety was monitored throughout the trial and follow-up (30 ± 3 days after last administration). Treatment-emergent adverse events (TEAEs) that started after DTIC subjects had crossed over to pimasertib were counted in the pimasertib (crossover) group. TEAEs were coded according to the most up-to-date version of MedDRA (version 18; protocol amendment September 2015) and graded by investigators according to National Cancer Institute Common Terminology Criteria for Adverse Events (NCI-CTCAE, version 4.0). The criteria for de-escalation or discontinuation of pimasertib are provided in Appendix A.

### 4.4. Statistical Analysis

Trial sample size was calculated based on median PFS with DTIC of 1.6 months for *BRAF* mutated metastatic melanoma and 2.8 months for unselected patients [17,18]. Assuming median PFS of 2 months under DTIC and a hazard ratio (HR) of 0.57 versus pimasertib, 151 observed events would provide 90% power to detect a difference at a two-sided, 5% significance level. Random (2:1) treatment assignment of 184 patients within 14 months should give a study duration of 34 months (assuming 10% loss to follow-up).

The primary endpoint was investigator-assessed PFS, defined as the duration from randomization to the first documentation of objective PD or death. PD or death were censored if they occurred after ≥2 missed tumor assessments. PFS was compared using a stratified log-rank test, with HR calculated by a Cox proportional hazards model stratified by baseline ECOG PS (0 versus 1). A sensitivity analysis of the primary endpoint was performed using independent central review data, with additional sensitivity analyses including all deaths and scans.

Secondary endpoints included: objective response rate (ORR); disease control rate (DCR; complete response (CR), partial response (PR) and stable disease (SD) for >3 months); PFS at 6 months from randomization; overall survival (OS); change in QoL; TEAEs (occurring within safety follow-up); serious AEs (SAEs); AEs of special interest (AESIs; including serous retinal detachment (SRD) and retinal vein occlusion (RVO), and creatinine phosphokinase (CPK)-related AEs); and deaths. Sensitivity analyses of ORR and DCR were conducted using independent central review data. P-values for secondary endpoints and sensitivity analyses were regarded as exploratory. The primary endpoint, PFS by independent review and OS were also analyzed according to prespecified baseline subgroups.

Efficacy and sensitivity analyses were performed using the intent-to-treat (ITT) population (patients randomized) and safety analyses were carried out in the safety population (patients who received ≥1 dose of study treatment). The cut-off date of the main analysis for the evaluation of primary and secondary objectives was 04 July 2015 (approximately 12 months after randomization of the last patient).

## 5. Conclusions

Pimasertib has shown clinical activity in patients with *NRAS*-mutated cutaneous melanoma and a safety profile that is consistent with the known toxicities of *MEK* inhibitors. These findings support the potential combination of pimasertib with other agents, such as those targeting the *PI3K*/*mTOR* pathway or other *MEK* pathway components, or immune checkpoint inhibitors.

## Figures and Tables

**Figure 1 cancers-12-01727-f001:**
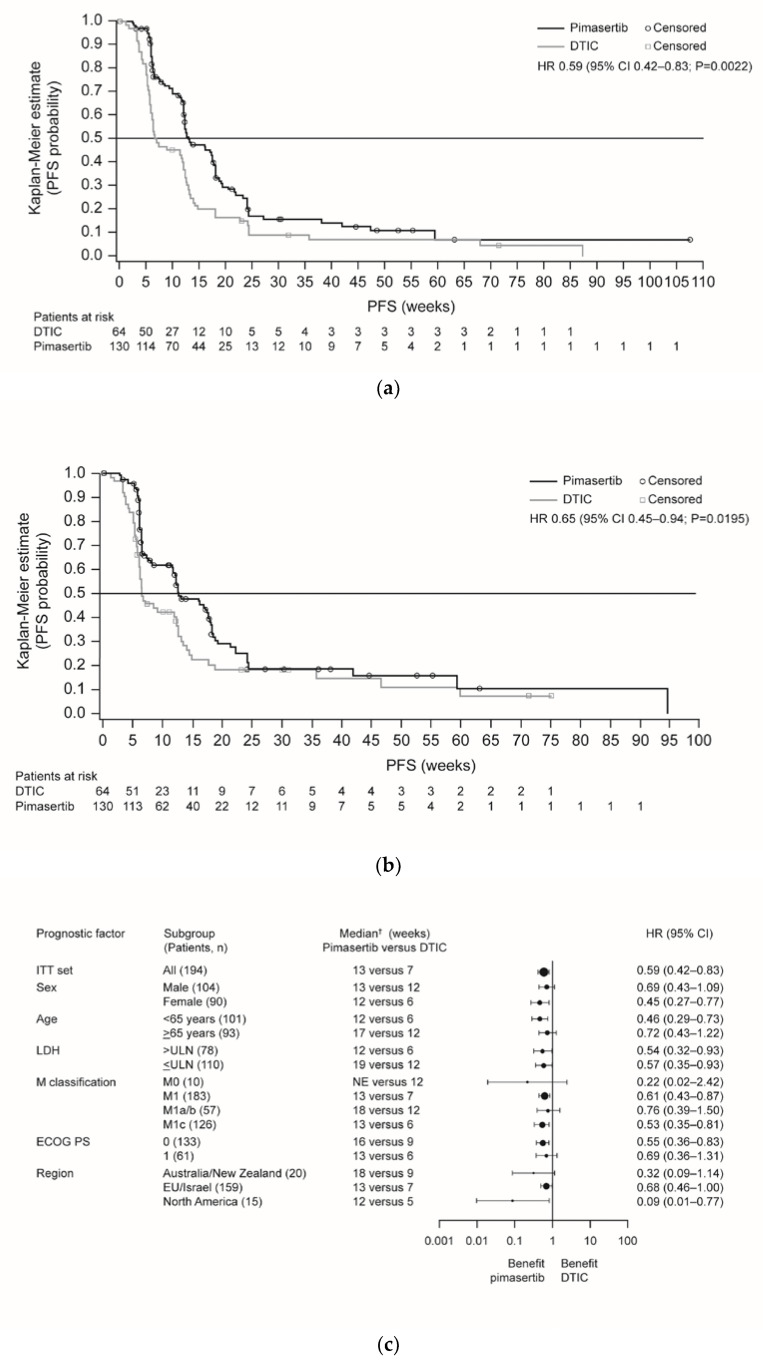
Kaplan-Meier plots of PFS for pimasertib versus DTIC based on endpoints: (**a**) read by investigator assessment (primary endpoint); (**b**) independent assessment of the data (sensitivity analysis); (**c**) forest plot of PFS for pimasertib versus DTIC based on investigator assessment (ITT analysis set). Abbreviations: CI, confidence interval; DTIC, dacarbazine; ECOG PS, European Cooperative Oncology Group performance status; EU, European Union; HR, hazard ratio; ITT, intent-to-treat; LDH, lactate dehydrogenase; NE, not evaluable; PD, progressive disease; PFS, progression-free survival; ULN, upper limit of normal.

**Table 1 cancers-12-01727-t001:** Patient demographics and baseline characteristics (ITT analysis set).

Characteristic ^1^	DTIC (*n* = 64)	Pimasertib (*n* = 130)
Male:female, *n* (%)	36 (56):28 (44)	68 (52):62 (48)
Region, *n* (%)		
Australia/New Zealand	8 (13)	12 (9)
Europe	51 (80)	108 (83)
North America	5 (8)	10 (8)
Median age (range), years	62 (23–83)	65 (21–83)
ECOG PS, n (%)		
0	44 (69)	89 (69)
1	20 (31)	41 (32)
LDH level, n (%) ^†^		
>ULN	23 (36)	55 (42)
≤ULN	36 (56)	74 (57)
Melanoma stage at study entry, n (%) ^‡^		
Locally advanced	16 (25)	28 (22)
Metastatic	48 (75)	101 (78)
M classification, n (%)		
M0	2 (3)	8 (6)
M1a	5 (8)	16 (12)
M1b	14 (22)	22 (17)
M1c	43 (67)	83 (64)
Missing	0	1 (1)

^1^ All data have been rounded up to the nearest whole number. ^†^ Data for 5 and 1 patients missing in the DTIC and pimasertib groups, respectively. ^‡^ Data for 1 patient missing in the pimasertib group. Abbreviations: DTIC, dacarbazine; ECOG PS, Eastern Cooperative Oncology Group performance status; ITT, intent-to-treat; LDH, lactate dehydrogenase; ULN, upper limit of normal.

**Table 2 cancers-12-01727-t002:** Best overall response, ORR and DCR based on investigator and independent evaluation of data (ITT analysis set) ^1^.

Tumor Response	Investigator Evaluated		Independent Centrally Evaluated ^†^	
	DTIC (*n* = 64)	Pimasertib (*n* = 130)	DTIC (*n* = 64)	Pimasertib (*n* = 130)
Best overall response				
CR, *n* (%)	3 (5)	4 (3)	1 (2)	2 (2)
PR, *n* (%)	6 (9)	31 (24)	8 (13)	28 (22)
SD, *n* (%)	1 (2)	8 (6)	18 (28)	44 (34)
PD, *n* (%)	46 (72)	55 (42)	30 (47)	44 (34)
NE, *n* (%)	8 (13)	32 (25)	7 (11)	12 (9)
ORR, % (exact 95% CI)	14 (7–25)	27 (20–35)	14 (7–25)	23 (16–31)
Odds ratio (95% CI)	2.24 (1.00–4.98)		1.83 (0.81–4.13)	
*p*-value	0.0453		0.1430	
DCR, % (exact 95% CI)	16 (8–27)	33 (25–42)	27 (16–39)	38 (29–47)
Odds ratio (95% CI)	2.65 (1.23–5.69)		1.68 (0.87–3.26)	
*p* -value	0.0106		0.1235	

^1^ All data have been rounded up to the nearest whole number, except for odds ratios and P-values. ^†^ Sensitivity analysis. Abbreviations: CI, confidence interval; CR, complete response; DCR, disease control rate; DTIC, dacarbazine; ITT, intent-to-treat; NE, not evaluable; ORR, objective response rate; PD, progressive disease; PR, partial response; SD, stable disease.

**Table 3 cancers-12-01727-t003:** Most common grade ≥3 TEAEs (>5% of patients in any treatment group) (safety analysis set) ^1^.

Patients with TEAE, *n* (%)	DTIC (*n* = 61) ^†^	Pimasertib (*n* = 130)	Pimasertib crossover (*n* = 41)
Number of patients with at least one event (grade ≥3)	25 (41)	111 (85)	36 (88)
Blood CPK increased	0	44 (34)	15 (37)
Hypertension	2 (3)	12 (9)	3 (7)
Dermatitis acneiform	0	9 (7)	4 (10)
Ejection fraction decreased	0	9 (7)	1 (2)
Diarrhea	0	8 (6)	3 (7)
Dyspnea	0	8 (6)	0
Anemia	3 (5)	5 (4)	3 (7)
General physical health deterioration	1 (2)	1 (1)	4 (10)
Thrombocytopenia	8 (13)	0	2 (5)
Neutropenia	9 (15)	0	0

^1^ All data have been rounded up to the nearest whole number. ^†^ Includes events reported on DTIC treatment until 33 days after the last dose of DTIC. Any events that started after patients had crossed over to pimasertib treatment (*n* = 41) are counted in the pimasertib crossover group. Abbreviations: CPK, creatine phosphokinase; DTIC, dacarbazine; TEAE, treatment-emergent adverse event.

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
