# Peer review of "Pimasertib Versus Dacarbazine in Patients With Unresectable NRAS-Mutated Cutaneous Melanoma: Phase II, Randomized, Controlled Trial with Crossover"

_cancers, 2020, doi:10.3390/cancers12071727_

Round 1
Reviewer 1 Report
The study by Lebbe et al reports the efficacy and safety of pimasertib (MEK1/MEK2 inhibitor) versus dacarbazine (DTIC) in patients with untreated NRAS-mutated melanoma. The manuscript presents all relevant and detailed data. The major pitfall of this study, as also mentioned by the Authors, is that a reference drug (DTIC) is not further recommended for patients with metastatic melanoma. This makes this manuscript less timely.
Specific comments:
1) Gene names should be written in italics.
2) lines 70-73: encorafenib (BRAFi) and binimetinib (MEKi) that have been approved by FDA should be mentioned additionally
3) lines 118-129 - what does it refer to?
4) Please clearly state in the manuscript how NRAS mutatonal status was determined.
Author Response
1) Gene names should be written in italics.
This was corrected (NRAS, BRAF, MEK, PI3K and mTOR)
2) lines 70-73: encorafenib (BRAFi) and binimetinib (MEKi) that have been approved by FDA should be mentioned additionally
This was done accordingly in the manuscript
3) lines 118-129 - what does it refer to?
We apologize; this sentence was part of figure 1 legend
4) Please clearly state in the manuscript how NRAS mutatonal status was determined.
For confirmation of entry criteria, centralized testing of N-Ras mutation status was to be done ona tumor sample using the Sanger method. If the N-Ras status was already known at time of inclusion in the trial, the method used for testing and the date were also to be recorded. N-Ras status had to be known prior to performing trial specific screening assessments and prior to randomization in the trial.Analysis of the presence of different types of genetic variants (e.g. gene mutations, copy numbervariations, methylations), expression levels pathway associated proteins, and marker ofproliferation were to be performed on fresh tumor tissue biopsies or archived tumor tissue.

Reviewer 2 Report
The authors present a very concise report of the results of a well-sized and powered phase 2 trial of MEK inhibition in advanced NRAS mutated melanoma. The data are presented well, and conclusions are appropriate. Notwithstanding the quality of presentation, the authors do note a significant limitation in that their control arm no longer represents standard-of-care therapy. Additionally, it is preceded by phase 3 data of analogous therapy with binimetinib, albeit published at an impressively immature time point of 1.7 months median follow-up. By contrast, it is a little surprising that this manuscript is emerging 5 years after the data cut-off date, but it is likely that changing treatment paradigms and novelty have factored in to this.
Table 1: the numbers for stage at diagnosis (locally advanced versus metastatic) and the M-stage breakdown immediately below them do not appear to be consistent (e.g. in the DTIC arm metastatic stage n=48, yet n=62 have an M stage of M1a/b/c; it's not clear how anyone could consider a patient with visceral metastasis to be "locally advanced"). Presumably the stage at diagnosis means stage at study entry? Please review these data for accuracy and remove ambiguity.
Lines 118-129 are supposed to be part of the legend for Figure 1 but has been incorporated directly into the text; please correct.
If 22-25% of patients had locally advanced unresectable disease (assuming the numbers are correct; see the question above), this presumably includes predominantly nodal and in-transit lesions. The use of RECIST v1.0 to assess radiographic responses in this specific cohort could have produced results that are somewhat different from what the reader might intuitively expect based on other and more recent targeted therapy data using RECIST v1.1 or the diversity of novel immune-related metrics. Admittedly RECIST v1.1 clarifications were introduced only shortly before patient recruitment began to this study, but there has been ample time to retrospectively review these results using other metrics.
The authors appropriately raise the issue of the control arm no longer representing standard-of-care. It would be beneficial if more details could be added (probably in the discussion) regarding the expected clinical outcomes in NRAS mutated melanoma patients receiving current first-line standards-of-care, particularly immunotherapy, including response rate and/or PFS data.
Author Response
The authors present a very concise report of the results of a well-sized and powered phase 2 trial of MEK inhibition in advanced NRAS mutated melanoma. The data are presented well, and conclusions are appropriate. Notwithstanding the quality of presentation, the authors do note a significant limitation in that their control arm no longer represents standard-of-care therapy. Additionally, it is preceded by phase 3 data of analogous therapy with binimetinib, albeit published at an impressively immature time point of 1.7 months median follow-up. By contrast, it is a little surprising that this manuscript is emerging 5 years after the data cut-off date, but it is likely that changing treatment paradigms and novelty have factored in to this.
Table 1: the numbers for stage at diagnosis (locally advanced versus metastatic) and the M-stage breakdown immediately below them do not appear to be consistent (e.g. in the DTIC arm metastatic stage n=48, yet n=62 have an M stage of M1a/b/c; it's not clear how anyone could consider a patient with visceral metastasis to be "locally advanced"). Presumably the stage at diagnosis means stage at study entry? Please review these data for accuracy and remove ambiguity.
- We would like to thank the reviewer for this comment; we confirm that in table 1, TNM was determined at study entry; we modified in table 1 ‘melanoma stage at diagnosis’ by ‘melanoma stage at study entry’.
Lines 118-129 are supposed to be part of the legend for Figure 1 but has been incorporated directly into the text; please correct.
- Lines 118-129 refer to the figure legend; we put the figure legend directly under the figure to facilitate the reading (L117-121). In addition, PFS definition was put directly in the assessment section in materiel and methods (section 4.3; L257-262) instead of in figure legend.
If 22-25% of patients had locally advanced unresectable disease (assuming the numbers are correct; see the question above), this presumably includes predominantly nodal and in-transit lesions. The use of RECIST v1.0 to assess radiographic responses in this specific cohort could have produced results that are somewhat different from what the reader might intuitively expect based on other and more recent targeted therapy data using RECIST v1.1 or the diversity of novel immune-related metrics. Admittedly RECIST v1.1 clarifications were introduced only shortly before patient recruitment began to this study, but there has been ample time to retrospectively review these results using other metrics.
- We would like to thank the reviewer for this comment; however, the Sponsor states that in the present study, RECIST 1.0 was used as standard at that time. The RECIST 1.1 was published in 2016, when the study was already fully enrolled. The study started in 2012 and completed in 2015. Unfortunately, the study team and the pimasertib development team no longer exist in the company and there is no longer any resource dedicated to pimasertib within Merck since clinical development has been stopped, and so reanalysis of data is not possible at that stage.
The authors appropriately raise the issue of the control arm no longer representing standard-of-care. It would be beneficial if more details could be added (probably in the discussion) regarding the expected clinical outcomes in NRAS mutated melanoma patients receiving current first-line standards-of-care, particularly immunotherapy, including response rate and/or PFS data.
- We would like to thank the reviewer for this comment; we added in the discussion references [24] and [25] concerning PD-1 blockade either alone or combined with anti-CTLA (Larkin J, 2015 and Robert C, 2019) in lines L216-220. We also adapted ‘References’ section in the manuscript (L401-405).